# Continuous Increase of Efficacy under Repetitive Injections of Botulinum Toxin Type/A beyond the First Treatment for Adult Spastic Foot Drop

**DOI:** 10.3390/toxins13070466

**Published:** 2021-07-02

**Authors:** Harald Hefter, Werner Nickels, Dietmar Rosenthal, Sara Samadzadeh, Philipp Albrecht

**Affiliations:** 1Department of Neurology, University Hospital, Moorenstrasse 5, D-40225 Düsseldorf, Germany; wesutoch@gmx.de (W.N.); rosenthal@med.uni-duesseldorf.de (D.R.); sara.samadzadeh@yahoo.com (S.S.); phil.albrecht@gmail.de (P.A.); 2Department of Neurology, Ruland-Kliniken, Neuenbürger Strasse 49, D-75335 Dobel, Germany

**Keywords:** spastic foot drop, functional benefit, gait velocity, active range of movement, aboBoNT/A, motor learning

## Abstract

The objective of this study was to quantify the increase in efficacy during the first four cycles of treatment with botulinum toxin type/A (BoNT/A) in 24 free-walking BoNT/A naïve adult patients with post-stroke hemispasticity and spastic foot drop. Patients were followed over 390 days and received five injections of 800 U aboBoNT/A every three months. Patients assessed the treatment effect at eight visits using a global assessment scale, physicians scored the muscle tone at the ankle joint, measured active and passive ranges of motion (aRoMs, pRoMs) at the knee and ankle joint and determined the distance patients succeeded to walk during a minute. Patients’ assessments significantly (*p* < 0.006) increased with time and significantly correlated with all parameters measured. The best correlation (r = 0.927; *p* < 0.0001) was found with the sum of the aRoMs of knee and ankle joint. After one year of treatment outcome measures were better than and significantly correlated with the peak effect of the first injection. This correlation was higher for pRoMs (r = 0.855; *p* < 0.00001) compared to aRoMs (r = 0.567; *p* < 0.009). When BoNT/A treatment of the spastic foot in chronic hemispasticity is performed regularly every three months for at least one year, patients will experience a significant increase of benefit beyond the first treatment, but have to learn how to adapt to and use the new degree of freedom induced by the injections.

## 1. Introduction

Stroke is among the most relevant causes of death and disability, and it affects up to 0.2% of the population in industrial countries [1]. If the patient survives the event, a broad spectrum of disabling symptoms can persist or appear where motor deficits are the most common type [2]. Between 19 to 43% of stroke survivors will develop spasticity [3,4,5,6] as a “plus”–symptom of the so-called Upper Motor Neuron Syndrome (UMNS) [7]. Further “plus”–symptoms of the UMNS include increased muscle tone, enhanced tendon reflexes, flexor, and/or extensor spasms as well as complex associated reactions and abnormal postures of trunk and limbs [7,8]. On the other hand, “minus”–motor symptoms include, loss of strength and dexterity [7]. Distribution of “plus”– and “minus”–symptoms depend on the localization and size of the lesion in the sensorimotor system [7]. Both “plus”– and “minus”–symptoms considerably handicap a patient with UMNS.

Intramuscular injections of botulinum neurotoxin typeA (BoNT/A) are highly effective in reducing muscle tone in spastic muscles and have received level A recommendations of the Therapeutics and Technology Assessment Committee of the American Academy of Neurology [9] and the Royal College of Physicians [10] for treatment of chronic adult spasticity.

Despite the significant reduction of muscle tone, the functional benefit has less consistently been demonstrated for BoNT/A-therapy of the lower limbs [11]. Patients may to be satisfied with BoNT/A therapy, even when a significant increase of gait velocity is not achieved [12]. There are positive [13,14,15,16,17,18] as well as negative [12,19,20,21] reports on an increase of gait speed after BoNT treatment. Even when BoNT treatment is combined with intensive physiotherapy gait speed may not necessarily be significantly increased [22].

However, when the effect of BoNT treatment of lower limbs is measured by means of an instrumental gait analysis a variety of different functional benefits can be detected even when clinical measures as gait speed do not improve [23]. This can be observed in chronic stroke survivors as well as in children with cerebral palsy (CP; for reviews see [23] and [24]). A consistent finding is improvement of ankle dorsiflexion throughout most of the gait cycle and an earlier knee flexion during swing phase [24]. This is consistent with the observation in 24 BoNT-naïve adult post-stroke patients that the sum of the active range of motion (aRoM) around the ankle and knee joint was improved after the first injection of 800 U abobotulinumtoxin type A (aboBoNT/A; Dysport^®^, IPSEN^®^, France) into the calf muscles in all patients.

For the improvement of spasticity of the upper extremity, the relevance of the increase of aRoMs has clearly be demonstrated [25,26,27]: “Increase in active range of motion and steady rest angle contributed most to prediction of functional outcome” [28]. Early measures of aRoMs could predict recovery of upper extremity function three months after stroke [29]. The relevance of aRoM measures for lower limb function can also be suspected but has less clearly been demonstrated. To underline the relevance of aRoM measures also in BoNT treatment of leg spasticity patient’s assessment of the efficacy of BoNT treatment is compared to detailed angle measurements in the present study.

In children quality of gait was improved by a single BoNT treatment [30]. With repeated injections every seven months, the effect of the first injection could be reproduced, but no further improvement beyond the effect of the first cycle was found [30]. Here we demonstrate that with repeated injections every three months the effect on adult lower limb spasticity will increase far beyond the peak effect of the first cycle. This is similar to BoNT treatment of focal cervical dystonia where injections every three months will lead to a mean improvement of baseline severity (before the next injection) of more than 50% [31,32].

## 2. Results

### 2.1. Demographical and Safety Data

The present clinical study included more male than female post-stroke patients and more patients with infarcts than patients with hemorrhages. The age at baseline visits and duration since stroke covered a wide range (Table 1). One of the patients experienced a serious adverse event (SAE) not related to medication during the first three months but could be analyzed thoroughly. During the second injection cycle, four patients were lost to follow-up: three patients had a SAE not related to medication (one death, two suffered a further stroke), one patient moved to another part of the country. The rest of the patients (*n* = 20) completed the entire study. Patients were mildly to severely affected and walked much slower than in other studies (as for example in [17]. During everyday life activities 10 patients wore orthopedic shoes, five patients wore an ankle-foot-orthosis (AFO), five patients were cane users and two patients used a wheel walker (see Table 1).

### 2.2. Analysis of the First Treatment Cycle

All 24 patients completed the first cycle. None of the 24 patients reported a negative effect of the first aboBoNT/A injection on spasticity or gait. 

At day 30, four patients reported no change (score = 0), 13 patients a mild improvement (score = 1) and seven patients a good improvement (score = 2). Mean patient’s global assessment (PGA) was significantly higher than zero (PGA: 1.125; SD: 0.68; *p* < 0.05; Figure 1A). During the next 60 days, PGA again declined, but remained close to 0.75 and significantly above 0 (*p* < 0.05).

All eight angles measured were significantly (*p* < 0.01 (after Bonferoni adjustment)) improved, with only one exception. ARoM at the ankle joint in flexion/extension direction did not change significantly (*p* = 0.16). The sum of knee (K)-aRoM and ankle (A)-aROM was highly significantly (*p* < 0.001) improved 30 and 60 days after the first injection (Figure 1B) and this effect only slightly diminished during the following 30 days. The coefficients of variation (VAR = S.D./MV) for aRoMs were at least two times larger than corresponding VARs of pRoMs. VARs of K+A-RoMs were among the lowest observed (baseline-VAR: K+A-aRoM = 50%, K+A-pRoM: 15%; 30 days-VAR: K+A-aRoM = 38%, K+A-pRoM = 11%).

Muscle tone (MAS) continuously declined during the first injection cycle (Figure 1C), but the differences to baseline were not significant.

The mean gait velocity at baseline was 0.292 m/s (WD/1 min = 17.52; S.D. = 12.12) and ranged from the extremely low value of 3 m/1 min (=0.05 m/s) to 40 m/1 min (=0.67 m/s) which is well below normal gait speed. Mean increase of gait speed after 30 days was small, but significant (1.63 m/1 min = 0.027 m/s; *p* < 0.036). During the next 60 days mean gait velocity further increased up to 0.368 (WD/1 min = 22.1; S.D. = 13.35; *p* < 0.018; Figure 1D). Rm-ANOVA did not reveal any influence of age, sex, and time since stroke on the outcome (PGA, MAS, WD/1 min, and all angle measurements) during the first injection cycle.

### 2.3. Temporal Development of Efficacy Parameters with Repetitive Treatment

With subsequent injections, PGA continued to increase up to a mean value of more than 1.5 (Figure 1A) at day 360. The correlation of mean PGA of the eight visits with time (days after the first injection) was significant (r = 0.863; *p* < 0.006). MAS continuously and highly significantly decreased (r = −0.966; *p* < 0.0001; Figure 1C), whereas WD/1 min slightly, but significantly increased (r = 0.738; *p* < 0.036; Figure 1D). 

K+A-aRoMs highly significantly increased (r = 0.883; *p* < 0.004; Figure 1B), whereas K+A-pRoMs only slightly, but significantly increased (r = 0.775; *p* < 0.024; diamonds in Figure 2B). The similarity between the temporal development of PGA and K+A-aRoMs is obvious (Figure 1A,B).

In Figure 2 the changes from baseline of aRoMs and pRoMs are compared. Open symbols indicate values at control visits at day 30 and 60 and 390, full symbols indicate values obtained at visits for the next BoNT/A injection at days 90, 180, 270, and 360. With repetitive injections, the increase of aRoMs over time (K-aRoM: 0.032 deg/day; K+A-aRoM: 0.090 deg/day; squares in Figure 2A,B) is steeper than the increase of pRoMs (K-pRoM: 0.014 deg/day; K+A-pRoM: 0.024 deg/day; diamonds in Figure 2A,B). Therefore, the difference between the larger pRoMs and the smaller aRoMs becomes smaller with repeated 3-monthly aboBoNT/A injections.

When mean values of the parameters measuring efficacy of aboBoNTA therapy were correlated (Table 2) the correlation coefficient between PGA and gait velocity was lower (r = 0.783, *p* < 0.022) than between PGA and MAS (r = 0.816; *p* < 0.014) and much lower than between PGA and K+A-aRoMs (r = 0.927; *p* < 0.001). The lowest correlation coefficient was found between gait velocity and MAS (r = −0.740; *p* < 0.036), the highest between K+A-aRoMs and K+A-pRoMs (r = 0.959; *p* < 0.0002).

### 2.4. Comparison of Data at the Baseline Visit (Day 0) with the Peak Effect of the First Injection (Day 30) and Comparison of the Peak Effect of the 5th Injection (Day 390) with the Baseline before the 5th Injection (Day 360)

Compared to the baseline values at day 0, all parameters presented in Table 3 were improved at day 390 (Table 3). The peak effect of the 5th injection (the difference between day 360 and day 390 (Table 3)) was always smaller than the peak effect of the 1st injection (the difference between day 0 and day 30 (Table 3)).

The improvement of the baseline values before the 5th injection after one year of repeated aboBoNTA injections every three months (the difference between day 0 and day 360 (Table 3)) was consistently much larger (except for K+A-pRoMs) than the peak effect of the 1st injection (day 30, Table 3)).

### 2.5. Correlation between the Peak Effect of the First Injection (Day 30) and the Baseline Assessment before the 5th Injection (Day 360) after One Year of Continuous Treatment

To test whether the peak effect of the 1st injection is of predictive value for the outcome after 1 year of continuous aboBoNT/A treatment, K+A-pRoMs (x-axis in Figure 3A) and K+A-aRoMs (x-axis of Figure 3B) at day 30 were compared to the corresponding data at day 360 (y-axis in Figure 3 A,B). K+A-aRoMs at day 360 were significantly larger than at day 30 (*p* < 0.001), but not K-A-pRoMs (*p* = 0.16; n.s). K+A-pRoMs at day 30 were highly significantly correlated with K+A-pRoMs at day 360 (r = 0.855; *p* < 0.00001; Figure 3A). K+A-aRoMs at day 30 were also significantly correlated with K+A-aRoMs at day 360 but to a less extent (r = 0.567; *p* < 0.009; Figure 3B).

## 3. Discussion

### 3.1. General Remarks on Inducing a Functional Improvement of Gait in Chronic Hemispasticity

A functional improvement of gait after BoNT therapy appears to be difficult to achieve in adult patients with chronic hemispasticity [11]. These patients shift their lower body to the less affected side [33] for better support of the center of mass. The upper body is tilted to the more affected side compensating the shift of the lower body [34,35]. BoNT injections into the affected arm reduce the moment of inertia of the upper body, reduce the tilt of the upper body to the affected side and therewith increase gait velocity [36]. Therefore, in studies aiming at the analysis of improvement of gait BoNT injections into the arm have to be controlled. In the present study, no BoNT injections into the affected arm had been performed.

BoNT injections reduce activity of extra- and intrafusal muscles [37,38], reduce enhancement of spinal reflexes [39] and Ia-afferent input to spinal circuits [40], but also reduce force production of the muscles injected. Reduction of enhanced reflexes will improve gait by influencing the spinal cord and spinal central pattern generators [41,42]; reduction of force production may impair gait by reducing push-off. Furthermore, the pattern of impairment of different descending (corticobulbar, corticospinal, and bulbospinal) pathways [7] may vary from patient to patient. Thus, it is difficult to predict on the basis of clinical data whether a patient will have a functional benefit or not after BoNT treatment. Therefore, an instrumental gait analysis (IGA) is helpful to demonstrate functional improvement of gait.

However, most IGA systems to not allow a detailed analysis of foot movements because of a too simple foot model. Furthermore, free-walking does not test patient’s full capacity of voluntary control of movements around a single joint. We,, therefore, have measured in detail the passive and active range of motion. Increase of aRoMs at the knee and angle joint could be demonstrated as well as an associated increase of gait speed. We have also used the Infotronic^®^ gait analysis system (NL-7650 AB Tubbergen, The Netherlands) measuring the foot pressure during walking. Similar to patients with ICP we were able to demonstrate an improvement of the area of foot pressure after BoNT/A injections into the affected leg and a decrease of gait asymmetry. These results will be reported separately.

### 3.2. General Remarks on Injecting Patients Every 3 Months

In the present study, free walking adult patients with a spastic foot drop experienced a continuous improvement with repeated botulinum toxin type A (800 U aboBoNT/A) injections every three months. Injection by injection the PGA indicated an increasing benefit of BoNT/A therapy from the patients’ perspective (Figure 1A). For all parameters measured a level of improvement was observed after one year of BoNT treatment which was higher than the peak effect of the first injection (Table 3). The clinical implication, therefore, is to reinject adult post-stroke patients with lower leg spasticity on a regular basis every three months.

This recommendation is supported by an observation in our BoNT outpatient clinic during the SARS COV2 pandemia induced lock-down in Germany spring 2020. About 80% of the patients in the BoNT ambulance were injected every three months. As a consequence of the lock-down reinjection had to be postponed for two to three days up to four weeks in more than 100 patients. Assessment of the worsening during this delay by patients and treating physicians revealed a worsening by 1%/1 day, summing up to a clinically highly relevant worsening of 30% after one month. This was observed across all indications (patients with focal dystonia, patients with spasticity, patients with pain syndromes; for details see [32]). This underlines the gradual benefit patients have when they are reinjected before the clinical effect of the previous injection has completely vanished.

The significant increase of gait speed of 0.027 m/s 30 days after the 1st injection was well below the value of 0.04 m/s which was found to be significant in the meta-analysis by Foley et al. [43] and below the non-significant increase of 0.043 m/s reported by Burbaud et al. [12] The significant improvement of 0.072 m/s observed in the present study after 360 days of continuous 800 U aboBoNTA treatment was well above this 0.04 m/s limit.

### 3.3. Comparison on the Development of Active and Passive Ranges of Motion after Repeated BoNT/A Injections

In the present study, aRoMs increased more than pRoMs with repeated injections. This was demonstrated for knee angles (Figure 2A) and even more clearly for the sum of ankle and knee angles (Figure 2B). Passive angles were the only parameters for which the increase after one year of continuous BoNT/A treatment did not exceed the peak effect of the first injection (Table 3). In contrast, the increase of aRoMs at day 360 was about twice as large as the peak effect of the 1st injection (Table 3). This implies that the difference between the larger pRoMs and the smaller aRoMs becomes smaller and smaller with repeated injections. The steep increase of aRoMs indicates an increasing better control of the center of mass.

Therefore, it does not surprise that patients’ assessments of the efficacy of BoNT/A injections further increased with the number of injections applied (r=0.862; *p* < 0.006) and significantly correlated (r = 0.927; *p* < 0.001) with aRoMs, much better than with gait velocity (r = 0.783; *p* < 0.022).

It is important to mention that a plateau effect, indicating that the best possible outcome was reached, only became evident for MAS and WD/1 min at day 270 and 90, respectively, while the other outcomes PGA and aRoM were still in the linear phase of increase even at day 360. This has important implications not only for the design of future efficacy trials but also for the counceling of patient during routine clinical practice.

Similarly, to the present study in lower limb spasticity, it has been reported in a large double-blind placebo-controlled study on upper limb spasticity [44] that aRoMs were improved after a single injection of aboBoNT/A in the finger, wrist, and elbow flexors. But in contrast to the present study which reveals an improvement of gait velocity in parallel to the increase of aRoMs no change in functional outcome (measured by the Modified Frenchay scale) was observed for the upper extremity [44]. But similar to the study by Beebe et al. demonstrating that the early measurement of the active range of motion predicts upper extremity function three months later [29] it may very well be that a functional improvement occurs with further increase of aRoMs with repetitive BoNT injections. These observations emphasize that “improvements in non-functional tests may also translate into walking changes and lower limb coordination” [23]. The analysis of the maximal range of motion reveals the capacity of voluntary movement control achieved by BoNT injections clearly, but it has to be taken into account that improvement of aRoM has to be trained and learnt [26,27,45], and may need time to develop.

### 3.4. Predictive Value of the First Injection on Long-Term Outcome

There was a highly significant (r = 0.855, *p* < 0.00001) correlation between the outcome 30 days after the first injection and the outcome after one year of continuous 3-monthly injection therapy for the pRoMs (Figure 3A). The corresponding correlation for the aRoMs was less significant (r = 0.567; *p* < 0.009). This means that the outcome of pRoMs after one year of continuous aboBoNT/A treatment can precisely be predicted from the effect of the first injection. For aRoMs this prediction is less precise. Post-stroke patients may have difficulties in motor learning [45]. We, therefore, recommend tocontinue BoNT/A therapy of lower leg spasticity for at least one year.

The high correlation between patients’ scores (PGAs) and aRoMs (Figure 1A,B and Table 2) strongly suggests that improvement of aRoMs is highly relevant for the patient. Improvement of pRoMs increases the possible range for voluntary movements, increase of aRoMs implies better control of the center of mass. Patients seem to be interested in increased stability during walking, and only in a second line in increased velocity during walking.

The sum of angles is usually not analyzed but may be highly sensitive to detect changes of BoNT injections into lower leg muscles [18]. The interplay between various angles (as the elevation angles: ankle dorsiflexion, knee flexion, hip flexion) has been studied in detail [41,46]. The human CPGs of walking seem to coordinate functional units and limb segment motion rather than single joint movements [41,42]. This is probably the reason why the improvement of the sum of ankle and knee joint reveals such a high correlation with patient’s assessment of the efficacy of BoNT injections every three months.

## 4. Conclusions and Implications for Clinical Practice

Repeated injections over one year of 800 U aboBoNT/A under EMG-guidance for the treatment of spastic foot drop led to a continuous reduction of muscle tone, a continuous increase of aRoMs, and a continuously increasing patient assessment of the efficacy of BoNT therapy. Therefore 800 U aboBoNT/A seem to be an appropriate dose when improvement of gait function is the primary aim of BoNT/A treatment for lower leg spasticity. Since the utilization of the new degree of freedom achieved by an increase of pRoMs after BoNT injection may need time until it is transformed into an increase of aRoMs, lower limb spasticity should be treated with BoNT/A repeatedly every three months for at least one year. The low variation of the sum of knee and ankle angles and the high correlation between PGA and K+A-aRoMs indicate that angle combinations may be more sensitive outcome measures than changes of angles at single joints.

## 5. Limitations of the Study

Patients’ assessment continuously increased with duration of BoNT/A therapy. Improvement of pain was not addressed, but may have considerably influenced [32] patient’s assessment of the efficacy of BoNT/A treatment. The problem of impaired motor learning has not been analyzed, but may be relevant to explain why some patients experience a better improvement of gait than others. With 24 patients, the present study has a rather small sample size. Furthermore, angles were measured during a sitting position. Measurement of angles during gait by means of an instrumental gait analysis will probably show the correlations between patient’s assessment and angle measurements more clearly but do not analyze the full capacity of motion achieved by BoNT injections.

## 6. Materials and Methods

This monocentric, open, prospective Investigator-Initiated Trial has been conducted in accordance with the Declaration of Helsinki. The local ethics committee has approved the investigation of treatment outcomes and treatment satisfaction at the BoNT outpatient clinic (Approval number 4085 (26 June 2017)) and all patients gave written informed consent.

### 6.1. Patient Sampling

Inclusion criteria were: (i) age ≥ 18 years; (ii) time since stroke ≥ 6 months; (iii) clinical evidence of absence of orthopedic or neurological deficits interfering with walking, other than stroke; (iv) no clinical indication for a brainstem, cerebellar, or ipsilateral hemispheric lesion; (v) ability to walk without aid for one minute without pauses; (vi) no previous BoNT-injection; (vii) patients without legal guardians for decision making or impaired judgement. Twenty-four free-walking BoNT-naïve adult patients with chronic hemispasticity after stroke, who were referred to our botulinum toxin clinic and met these inclusion criteria, gave informed consent, and were consecutively recruited.

### 6.2. Study Design

The effect of the first five consecutive aboBoNT/A injections was analyzed. Data recording (clinical investigation, assessment of treatment effect by the patient, scoring of muscle tone, angle measurements, analysis of gait velocity) was performed eight times: at the baseline visit (day 0), at day 30, 60, 90, 180, 270, 360, and day 390. Injections were performed at day 0, 90, 180, 270, and 360. A tolerance of plus/minus seven days was allowed for each visit.

### 6.3. Injections of 800 MU aboBoNT/A

BoNT injections were performed in the clinical routine and according to our internal standard for treatment of lower limb spasticity. In general, a total dose of 800 U aboBoNT/A was diluted in 4 mL saline. All patients received at least four and up to eight different injection sites with at least 100 U and up to 200 U aboBoNT/A per site. All patients received at least 500 U in the triceps surae muscle (soleus and/or gastrocnemius muscles. [16]) Additional 300 U could be distributed into other muscles as the tibialis posterior, extensor hallucis, abductor hallucis, flexor digitorum brevis, or flexor digitorum longus muscle or injected also into the triceps surae. Patients did not receive BoNT injections into the arm.

BoNT/A injections were performed under electromyography guidance using a Teflon coated 27-G 4-cm needle (Neuroline Inoject^®^; Ambu A/S; Baltorpbakken 13; DK-2750 Ballerup, Denmark).

### 6.4. Assessment of the Clinical Effect of aboBoNT/A Injections

The primary outcome measure of the present study was patient’s global assessment (PGA) of the BoNT/A treatment effect on gait. From day 30 on patients rated the clinical effect of the injection by means of a 7-point global assessment Likert-scale: (PGA: −3 = much worse; −2 = worse; −1 = slightly worse; 0 = unchanged; 1 = slightly better; 2 = better; 3 = much better) at each control visit.

An additional secondary (functional) outcome measure in the present study was the distance which the patient succeeded to reach during a 1 min, straightforward walk (walking distance, WD in meters). Gait velocity was calculated as WD/1 min. Patients had to walk barefoot or with shoes and/or an ankle-foot orthosis, but without a cane or wheel walker.

Muscle tone at ankle joint (A) in flexion/extension direction was scored as further outcome measure using the modified Ashworth scale (MAS; for data analysis, a value of 1.5 was assigned to a MAS score of 1+ as in previous studies [16,17,21]).

For angle measurements of the foot (A: flexion/extension direction) and the knee (K: flexion/extension direction), patients were comfortably seated with upper legs being supported, but with lower legs hanging down passively. For the passive range of motion (pRoM), the investigator examined how far he could move the foot or lower leg in a selected direction, for the active range of movement (aRoM), how far the patient could move the foot or lower leg in a selected direction. Four angles were determined (Knee: K-aRoM, K-pRoM, Ankle: A-aRoM, A-pRoM) and two angle combinations (K+A-aRoM = K-aRoM+A-aRoM; K+A-pRoM = K-pRoM+A-pRoM).

An Infotronic^®^ gait analysis system (NL-7650 AB Tubbergen) was used to measure foot pressure during walking additionally to compare the length of foot pressure on the ground (ground reaction forces) of the affected and non-affected leg to quantify gait asymmetry. Methodological details and results of the analysis of GRFs will be reported separately.

### 6.5. Statistics

All 24 patients completed the first injection cycle; four patients were lost during the second cycle (for further details see first paragraph of the result section). A repeated measurement ANOVA with posthoc Helmert contrasts was applied to PGA, MAS, WD/1 min, and the eight angle measurements (see above). The Bonferroni-Holm procedure was applied to the test statistics to fix the global level of significance to alpha = 0.05. An exploratory correlation analysis using Spearman’s rho was performed to determine the level of agreement between PGA, muscle tone, gait speed, and K+A-pRoMs and K+A-aRoMs, and the correlation of these parameters with time. The ANOVA and rank correlation are part of the commercially available SPSS^®^ statistical software package (version 25, IBM Corp., Armonk, NY, USA).

## Figures and Tables

**Figure 1 toxins-13-00466-f001:**
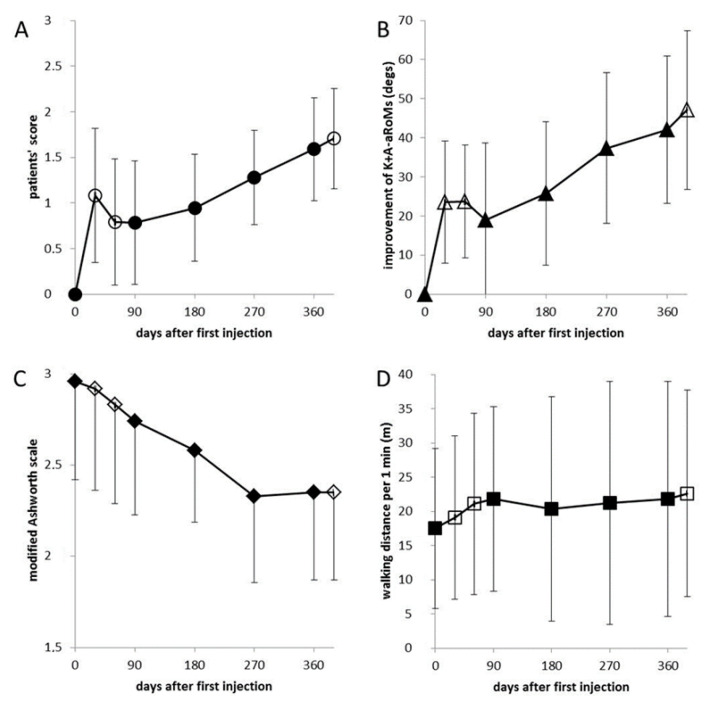
Mean values and standard deviations of patients’ assessment scores (PGA) (**A**), improvement of the sum of ankle and knee aRoMs (K+A-aRoMs) (**B**), modified Ashworth score (MAS) (**C**), and walking distance per 1 min (WD/1 min) (**D**) for all 8 visits are presented. All four parameters were significantly correlated with time (days after first injection): PGA: r = 0.863, *p* < 0.006; K+A-aRoMs: r = 0.883; *p* < 0.004; MAS: r = −0.966, *p* < 0.0001; WD/1 min: r = 0.738, *p* < 0.036. The similarity between the temporal development of patients’ scores and the aRoMs is quite apparent. (Open symbols represent visits without aboBoNT/A injections and full symbol visits at which patients received injections after the examination.). The first cycle was completed by all patients (day 0–90). Data of visits 5–8 (days 180–390) are based on 20 patients.

**Figure 2 toxins-13-00466-f002:**
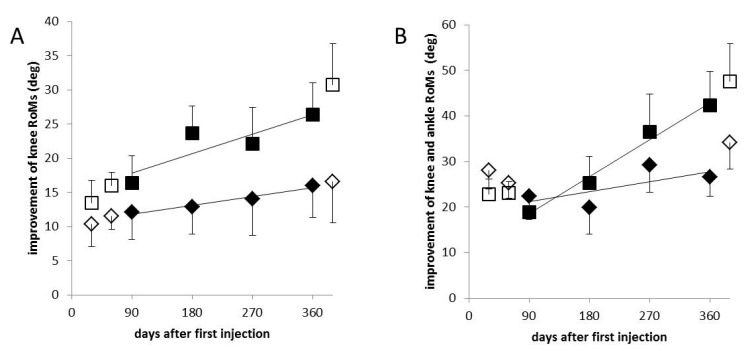
(**A**): mean values and S.D.s of the differences of K-pRoMs (diamonds) and K-aRoMs (squares) at visits 2–8 to the baseline visit are presented. The increase of aRoMs with repetitive injections is steeper than that of the pRoMs. (**B**): mean values and S.D.s of the differences of K+A-pRoMs (diamonds) and K+A-aRoMs (squares) at visits 2–8 to the baseline visit are presented. K+A-pRoMs (diamonds) only slightly, but significantly increased (r = 0.775; *p* < 0.024). The increase of aRoMs with repetitive injections is steeper than that of the pRoMs. (Open symbols represent visits without abo-BoNT/A injections and full symbol visits at which patients received injections after the examination).

**Figure 3 toxins-13-00466-f003:**
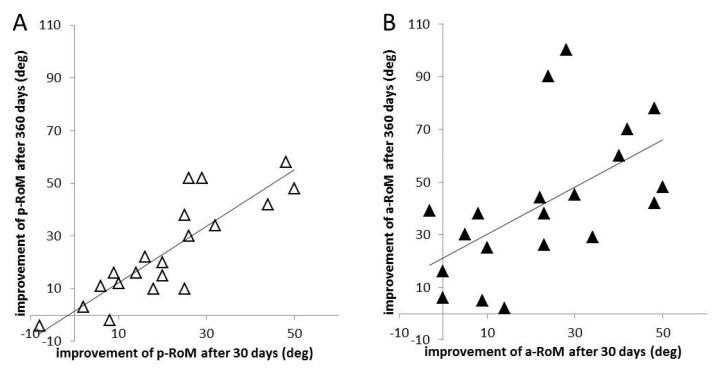
(**A**): Correlation between the differences of the sum of ankle and knee pRoMs at day 30 (at the time of the peak effect of the first injection) and the baseline visit (*x*-axis) and the differences of the sum of ankle and knee pRoMs at day 360 (just before the 5th injection was applied) to the baseline visit (*y*-axis). The correlation is highly significant (r = 0.855; *p* < 0.00001). (**B**): Correlation between the differences of the sum of ankle and knee aRoMs at day 30 (at the time of the peak effect of the first injection) and the baseline visit (*x*-axis) and the differences of the sum of ankle and knee aRoMs at day 360 (just before the 5th injection was applied) to the baseline visit (*y*-axis). The correlation is significant (r = 0.567; *p* < 0.009), but weaker than the correlation of the corresponding pRoM values (Figure 3A).

**Table 1 toxins-13-00466-t001:** Patients’ demographical data at baseline.

Age	MV: 54.25	range:
	S.D.: 12.93 years	22–77 years
duration since stroke	MV: 42.42	range:
	S.D.: 39.39 months	6–148 months
sex (male/female)	16 males	8 females
infarct/hemorrhage	16 infarcts	8 hemorrhages
walking aids	2 patients used orthopedic shoes only	
	3 patients used orthopedic shoes plus a cane	
	3 patients used orthopedic shoes plus an ankle-foot-orthosis	
	2 patients used orthopedic shoes, an AFO, and a cane	
	2 patients used a wheel walker	
arm injections	all 24 patients did not receive BoNT-injections into the arm	

MV = mean value; S.D. = standard deviation; AFO = ankle-foot-orthosis.

**Table 2 toxins-13-00466-t002:** Correlation coefficients (upper part) and *p*-values (lower part of the table).

Parameter	PGA	MAS	WD/1 min	K+A-pRoM	K+A-aRoM
PGA	-----	r = −0.816	r = 0.783	r = 0.927	r = 0.927
MAS	*p* < 0.014	-----	r = −0.740	r = −0.778	r =− 0.866
WD/1 min	*p* < 0.022	*p* < 0.036	-----	r = 0.909	r =−0.865
K+A-pRoM	*p* < 0.001	*p* < 0.03	*p* < 0.002	-----	r = 0.959
K+A-aRoM	*p* < 0.001	*p* < 0.001	*p* < 0.006	*p* < 0.0002	-----

PGA: patient’s global assessment scale; MAS: modified Ashworth scale; WD/1 min: walking distance per one minute; K+A-pRoMs: knee plus ankle passive range of movement; K+A-aRoMs: knee plus ankle active range of movement; r = Spearman’s rho.

**Table 3 toxins-13-00466-t003:** Changes from baseline to day 30, 360, and 390.

Parameter	PGA	MAS	WD/1 min	K+A-pRoM	K+A-aRoM
^a^ Diff 0 to 390	1.706	−0.61	5.12 = 0.085 m/s	34.77 (deg)	47.1 (deg)
^b^ Diff 360 to 390	0.117	0.0	0.83 = 0.014 m/s	6.25 (deg)	5.04 (deg)
^c^ Diff 0 to 30	1.125	−0.04	1.63 = 0.027 m/s	28.5 (deg)	23.27 (deg)
^d^ Diff 0 to 360	1.589	−0.61	4.29 = 0.072 m/s	27.1 (deg)	42.06 (deg)

PGA: patient’s global assessment scale; MAS: modified Ashworth scale; WD/1 min: walking distance per one minute; K+A-pRoMs: knee plus ankle passive range of movement; K+A-aRoM: knee plus ankle active range of movement. ^a^ Diff 0 to 390: change from baseline after 390 days of aboBoNT/A treatment, ^b^ Diff 360 to 390: peak effect of the 5th injection. ^c^ Diff 0 to 30: peak effect of the 1st injection. ^d^ Diff 0 to 360: change of baseline from day 0 to day 360 just prior to the 5th injection.

## Data Availability

Data available on request due to restrictions eg privacy or ethical. The data presented in this study are available on request from the corresponding author.

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
