# Peer review of "Continuous Increase of Efficacy under Repetitive Injections of Botulinum Toxin Type/A beyond the First Treatment for Adult Spastic Foot Drop"

_toxins, 2021, doi:10.3390/toxins13070466_

Round 1

Reviewer 1 Report

This study is evaluating he effect of repeat BoNT/A injections on post-stroke lower extremity spasticity by measuring patient impression, gait velocity, and passive and active ROM of the ankles and knees.  They found that each of these measures improved and showed continued improvement with subsequent injections.  However, there are several things that need to be addressed.  

  1. The authors refer to improvements in gait function, but the only measure of gait is gait velocity.  Increase in gait velocity is not synonymous with improvement in gait function.  The authors even state "increase of gait velocity may be associated with a higher risk of falls" which would be a negative outcome.  The authors need to be specific and consistent throughout the manuscript that they are measuring gait velocity.  Additionally, they need to further discuss that this is a single component of gait and does not fully assess gait function.
  2. The authors do not explain their reasoning of why they chose to measure ROM and how this may impact gait function.  They need to explain this and provide references for their reasoning, for example any correlation between improvement in ROM and measures of gait function such as fall reduction, etc.
  3. The authors define abbreviations for ROM and then do not use them consistently which creates confusion and makes the results difficult to follow and interpret.  They need to be consistent in the use of abbreviations throughout the manuscript.
  4. The grammar and punctuation need to be improved throughout the manuscript.  Some paragraphs (line 67-82 for example) are particularly difficult to follow.

Author Response

Reviewer 1 is absolutely right:

The present paper exclusively deals with clinical scores and measures. This is emphasized now.

Of course, increase of gait velocity is not synonymous with improvement in gait function.

However, we also used an easy-to-handle instrumental gait analysis system so that we know that gait was improved.  Results on this analysis will be published separately.

We now try our best to stick to the present clinical results and discuss these in relation to current literature.

Based on current reviews and other literature we emphasize the relevance of aRoMs.

Falls is a difficult topic. A recent Cochraine study did not mention that BoNT injections have influence on falls. We therefore only cite current literature how improvement of gait parameters may contribute to fall reduction.   

We are now consistent in the use of abbreviations throughout the manuscript.

We have improved grammar and punction. The automatic spell checking of word has been used. We have modified the text between line 67 and 82.

Reviewer 2 Report

This is a limited study about effect of Botulinum toxin for spasticity in foot .

The topic is interesting, but the manuscript should been improved.

My comments :

  1. Introduction: The section 2 about Botulinum toxin is not necessary, the readers know these facts already
  2. Material and methods: it remains unknown how pronounced were clinical findings
  3. How was spasticity in other parts than foot treated?
  4. The authors state that 4 patients were lost in the second cycle: What happened after second cycle- were additional patients lost. The reasons of loss?
  5. Because there are different number of patients in the cycles, the Figure 1 is not correct. The number of patients analyzed should be displayed
  6. The authors claim that improvement continues, but Figure 1C and Figure 1D supports not the fact
  7. Even though few patients in this study has been limitations, the additional limitations would be other spasticity treatment or  clinical findings in other spastic parts of body not showed in this manuscript

Author Response

We have shortened this paragraph, but think that there should be at least 1 sentence on the level A recommendation for BoNT treatment of spasticity.

Reviewer 2 is right:

We have added more clinical information in Tab. 1, especially on the severity of arm spasticity.

Details why patients were lost was presented in the result section. This is also mentioned in the statistics section and in Fig. 1 now.

In the first cycle 24 patients are analysed. In all other cycles 20 patients.

More detailed clinical information is presented now.

Reviewer 3 Report

In this study, objective of authors was to investigate and quantify the increase in efficacy beyond the first injection of botulinum toxin type/A in 24 naïve patients receiving repetitive 6 treatment for spastic foot drop.  They concluded repeated injections of 800 U led to a continuous reduction of muscle tone and a continuous increase of gait velocity. My comments:

1) Introduction and Discussion sections must be improved increasing comparison with other studies;

2) References must be improved increasing recent bybliography;

3) English requires moderate changes 

Author Response

Reviewer 3 is right:

We have added several recent studies on BoNT treatment of lower limb spasticity and instrumental gait analysis.

We now emphasize that more longitudinal studies are recommended.

Round 2

Reviewer 1 Report

The authors have made some improvements, but there are still issues remaining.  They still need to define what they are considering improvement in gait function and how the components they are measuring relate to gait.  They have not done this adequately.  They need to make additional improvements in grammar and punctuation.  In particular, many sentences need commas.

Author Response

Following the argument of reviewer 1 that gait speed is a very special aspect of gait function we have already changed the title from

“Gradually increasing functional improvement of the adult spastic foot drop under repetitive botulinumtoxin/A injections”

to

“Continuous increase of efficacy under repetitive injections of botulinumtoxin/A beyond the first treatment for adult spastic foot drop”.

Using detailed angle measurements we have demonstrated that the active range of motion at the knee and angle joint gradually increases with repeated injections. This increase of the passive and active range of movement is correlated with patients´ assessments of the efficacy of BoNT/A injections and gait speed.

We think that angle measurements (especially the measurement of the maximal active range of movement) are an adequate method to demonstrate a functional improvement after BoNT treatment, which is associated with an increase gait speed.

But to convince reviewer 1 that we have also controlled improvement of gait function in detail directly we have added a paragraph in the discussion and a sentence in the methods that the foot pressure on the ground (ground reaction forces; GRFs) and gait asymmetry was analysed additionally and an improvement was found. The analysis of GRFs and asymmetry will be reported in a subsequent paper (which is already ready for submission and deals with aspects of foot control which exceed the scope of the present paper completely).    

We also enhanced the grammar and punctuations.

Reviewer 2 Report

I have read a new version of the manuscript as well as the responses. I agree that the article may be published now

Author Response

Thank you so much for your time and helpful comments

Round 3

Reviewer 1 Report

The authors have made appropriate changes.